# Current and Future Therapies for Immunogenic Cell Death and Related Molecules to Potentially Cure Primary Breast Cancer

**DOI:** 10.3390/cancers13194756

**Published:** 2021-09-23

**Authors:** Ryungsa Kim, Takanori Kin

**Affiliations:** 1Department of Breast Surgery, Hiroshima Mark Clinic, 1-4-3F, 2-Chome Ohte-machi, Naka-ku, Hiroshima 730-0051, Japan; 2Department of Breast Surgery, Hiroshima City Hospital, 7-33, Moto-machi, Naka-ku, Hiroshima 730-8518, Japan; ymj5014266@gmail.com

**Keywords:** breast cancer, adjuvant therapy, neoadjuvant therapy, immunogenic cell death, damage-associated molecular patterns, antitumor immunity

## Abstract

**Simple Summary:**

How a cure for primary breast cancer after (neo)adjuvant therapy can be achieved at the molecular level remains unclear. Immune activation by anticancer drugs may contribute to the eradication of residual tumor cells by postoperative (neo)adjuvant chemotherapy. In addition, chemotherapy-induced immunogenic cell death (ICD) may result in long-term immune activation by memory effector T cells, leading to the curing of primary breast cancer. In this review, we discuss the molecular mechanisms by which anticancer drugs induce ICD and immunogenic modifications for antitumor immunity and targeted therapy against damage-associated molecular patterns. Our aim was to gain a better understanding of how to eradicate residual tumor cells treated with anticancer drugs and cure primary breast cancer by enhancing antitumor immunity with immune checkpoint inhibitors and vaccines.

**Abstract:**

How primary breast cancer can be cured after (neo)adjuvant therapy remains unclear at the molecular level. Immune activation by anticancer agents may contribute to residual tumor cell eradication with postsurgical (neo)adjuvant chemotherapy. Chemotherapy-induced immunogenic cell death (ICD) may result in long-term immune activation with memory effector T cells, leading to a primary breast cancer cure. Anthracycline and taxane treatments cause ICD and immunogenic modulations, resulting in the activation of antitumor immunity through damage-associated molecular patterns (DAMPs), such as adenosine triphosphate, calreticulin, high mobility group box 1, heat shock proteins 70/90, and annexin A1. This response may eradicate residual tumor cells after surgical treatment. Although DAMP release is also implicated in tumor progression, metastasis, and drug resistance, thereby representing a double-edged sword, robust immune activation by anticancer agents and the subsequent acquisition of long-term antitumor immune memory can be essential components of the primary breast cancer cure. This review discusses the molecular mechanisms by which anticancer drugs induce ICD and immunogenic modifications for antitumor immunity and targeted anti-DAMP therapy. Our aim was to improve the understanding of how to eradicate residual tumor cells treated with anticancer drugs and cure primary breast cancer by enhancing antitumor immunity with immune checkpoint inhibitors and vaccines.

## 1. Introduction

In the last two decades, the molecular mechanisms by which anticancer agents induce different types of cell death, including autophagy, apoptosis, and necrosis, have been elucidated [1]. Anticancer agent-induced cell death has therapeutic effects against various forms of cancer. The induction of a large amount of such cell death can result in marked tumor shrinkage, elicit antitumor immunity, and generate tumor-specific immunity with long-term immunological memory, thereby leading to a cancer cure [2]. Anticancer agent-induced immunity has been proposed as a model of “danger” signaling that leads to proinflammatory cytokine production [3]. Calreticulin (CRT) is an important initiator of this signaling; it contains tumor antigens (TAs) and tumor-associated antigens (TAAs) in dying cancer cells, and is exposed on the membrane surface and engulfed by immature and mature dendritic cells (DCs). Priming with DCs begins the process and prompts the production of cytotoxic T lymphocytes (CTLs) [4]. This type of cell death is classified as immunogenic cell death (ICD) [4]. ICD is caused by the release of damage-associated molecular patterns (DAMPs), such as adenosine triphosphate (ATP), CRT, high mobility group box 1 (HMGB1), annexin A1, and heat shock proteins (HSPs) 70/90, prior to apoptosis [5,6]. Although the anticancer agent-induced release of these DAMPs is associated with the induction of antitumor immunity, research has shown that DAMPs such as HMGB1, CRT, and ATP are also involved in tumor progression [7], metastasis [8], and drug resistance [9], thereby representing a double-edged sword [6].

Commonly used standard treatments for primary breast cancer consist of neoadjuvant chemotherapy (NAC) and adjuvant chemotherapy with anthracyclines (e.g., doxorubicin, epirubicin, and cyclophosphamide) and taxanes (e.g., paclitaxel and docetaxel), and radiation after surgical treatment. The achievement of pathological complete response (pCR) is a favorable prognostic factor for human epidermal growth factor receptor 2 (HER-2)-positive, triple-negative (TN), and high-grade hormone receptor (HR)-positive HER-2-negative breast cancers [10]. The presence and abundance of tumor-infiltrating lymphocytes (TILs) before NAC play crucial roles in the induction of therapeutic effects and favorable prognostic outcomes after chemotherapy for HER-2-positive and TN breast cancers [11]. After NAC, the activation of antitumor immunity through innate and adaptive immune responses, such as natural killer (NK) and CD8+ T cell activity, and the downregulation of immunosuppressive factors such as regulatory T cells (Tregs) and cytotoxic T lymphocyte antigen 4 within them, are important for the enhancement of the therapeutic effect [12,13]. However, even after the achievement of pCR after NAC and additional treatment targeting residual tumor cells, distant recurrence occurs in some patients. In addition, with adjuvant therapy, the means by which antitumor immunity is elicited after anticancer drug administration and the immune memory is maintained by T cells remain unclear. This review discusses immune activation by anticancer agents and the subsequent acquisition of antitumor immunity from the perspective of ICD, and describes therapies targeting the molecules related to tumor progression and anticancer drug resistance for the curing of primary breast cancer.

## 2. Cell Death Modalities and Immunogenicity

Various anticancer agents induce cellular modalities, including apoptosis, autophagy, and necrosis [14]. Apoptosis is generally associated with cell turnover under physiological conditions. In this process, macrophages recognize apoptotic cells as “eat me” signals, exposing their membranes to phosphatidylserine (PS) for cleaning and secreting anti-inflammatory cytokines such as interleukin (IL)-10 and transforming growth factor β (TGF-β) [15]. Many factors associated with this process must be considered. The exposure of apoptotic cell membranes to PS induces immune tolerance that prevents autoimmune responses to self-antigens, which is determined by nonimmunogenic apoptosis [15]. On the other hand, the membranes of tumor cells killed by anticancer agents are exposed to CRT before PS is externalized, followed by the release of DAMPs (e.g., HMGB1, ATP, and annexin A1) and proinflammatory cytokines (e.g., IL-1β, IL-6, and tumor necrosis factor-α (TNF-α)) from immature DCs, and finally the engulfment of TAs and TAAs released from dying tumor cells by immature DCs. The immature DCs are then activated to present TAs and TAAs to CD8+ T cells, resulting in the production of CTLs and a specific immune response that kills the tumor cells (Figure 1).

In general, necrosis produces proinflammatory cytokines, but is not necessarily linked to immune activation. When macrophages fail to eliminate apoptotic cells efficiently and promptly, secondary necrosis occurs, and the integrity of the cell membrane is compromised [16]. This compromise causes DAMP release from the cytoplasm or exposure to the cell membrane, stimulating an immune response and regulating necrosis and other necrotic processes, such as necroptosis [16]. Autophagy is a cellular defense mechanism that sequesters microorganisms by forming autophagosomes and removes pathogens from the cell by catabolizing them with lysosomes [17]. These processes are involved in innate and adaptive immune responses. The innate immune response is mediated by pathogen recognition receptors (PRRs), such as toll-like receptors (TLRs) and nucleotide-binding oligomerization domain-like receptors, which promote cytokine secretion, phagocytosis, and the activation of NK and NKT cells through signaling pathways mediated by nuclear factor-κB (NF-κB) [18]. The autophagic response plays critical roles in T cell function, differentiation, and homeostasis and in B cell development and survival [19].

## 3. Immunomodulation by Conventional Anticancer Agents

Anticancer drugs, such as anthracyclines and taxanes, are used commonly in the treatment of primary breast cancer. Preclinical and clinical studies have shown that anticancer agents induce immunomodulation in breast cancer cells. The mechanisms of action and immunomodulatory effects of these agents are summarized in Table 1.

### 3.1. Anthracyclines

The administration of doxorubicin induces interferon γ (IFN-γ) production by CTLs, which influences the therapeutic effect, and IL-17 production by γδ T cells, which promoted CTL accumulation in the tumor bed in a mouse model of breast cancer, suggesting that γδ T cells play an important role in doxorubicin-induced antitumor immunity [20]. Similarly, another mouse model of breast cancer showed that the therapeutic effect of doxorubicin depended on IFN-γ production by CD8+ T cells and that IL-17 and IL-1β induction was required to achieve a therapeutic effect [21]. The presence of γδ T cells, but not NK cells, is also important for therapeutic efficacy. These results suggest that anthracyclines promote TA-specific CD8+ T cell proliferation in tumor-draining lymph nodes (TDLNs) and the tumor infiltration of IL-17-producing γδ T cells, followed by the infiltration of activated IFN-γ-producing CD8+ T cells. Doxorubicin reduced myeloid-derived suppressor cell (MDSC) expression in the spleen, blood, and tumor bed, with a loss of residual MDSC function, in a mouse model of breast cancer [22]. Importantly, doxorubicin increased the numbers of CD4+ and CD8+ T cells, effector T cells, and NK cells and the expression of IFN-γ, granzyme B, and perforin in tumor-bearing mice [22]. Furthermore, MDSCs isolated from patients with breast cancer were sensitive to doxorubicin-induced cytotoxicity in vitro [22].

### 3.2. Taxanes

Taxanes, such as paclitaxel and docetaxel, bind to microtubules with high affinity and inhibit the function of mitotic spindles, resulting in mitotic arrest and, often, cell death associated with a non-apoptotic mitotic catastrophe [44]. In addition to being cytotoxic, taxanes affect the immune system, but this action is due to their immunostimulatory effect on cancer cells, and not simply to the inhibition of cell division. In patients with breast cancer, the administration of paclitaxel or docetaxel increases the serum levels of IL-2, IL-6, IFN-γ, and granulocyte macrophage colony-stimulating factor (GM-CSF) and peripheral NK cell activity, whereas NAC decreases the levels of IL-1 and TNF-α [32]. Paclitaxel reduced CD4+Foxp3+ Tregs independently of TLR4 signaling, and impaired cell viability and cytokine production of Tregs, but not CD4+Foxp3- effector cells [33]. Paclitaxel made tumor cells more sensitive to CTL-mediated cytotoxicity in a mouse model of breast cancer by increasing the permeability of granzyme B by perforin-independent CTLs [34]. This bystander effect allows a small number of CTLs to enhance the antitumor effect on cells that do not express a specific TA.

Nanoparticle albumin-bound (nab)-paclitaxel is a solvent-free formulation that enables the delivery of paclitaxel as a suspension of albumin nanoparticles. It can increase the delivery of albumin to a tumor by transporting it through a receptor; this process is known as transcytosis [45]. The immunomodulatory effects of nab-paclitaxel on breast cancer cells are similar to those of paclitaxel.

Docetaxel has been used to treat anthracycline-refractory breast cancer, reducing the percentage of MDSCs in the spleens of mice with mammary tumors and increasing selective CTL responses [35]. Furthermore, docetaxel treatment induced MDSCs to change from an M2-like phenotype to an M1-like phenotype, and elevated macrophage differentiation markers [35]. These immunomodulatory effects of docetaxel are due to its inhibition of signal transducer and activator of transcription 3 (STAT3) in MDSCs, which results in their selective differentiation into an M1-like phenotype [35]. STAT3 is required for the biological function of IL-10, and its inhibition restores the expression of proinflammatory cytokines such as IL-12 and TNF-α, leading to antitumor immunity [46].

### 3.3. Cyclophosphamide

Cyclophosphamide is an alkylating agent that inhibits protein synthesis by blocking the transcription of DNA to RNA, and it has immunosuppressive and immunomodulatory effects. The dosing schedule and dosage of this drug are important for the achievement of immunological effects. Cyclophosphamide can completely eradicate cancer cells at high doses and is relatively selective for T cells at low doses. A single low dose of cyclophosphamide selectively suppresses Tregs [23]. The metronomic administration of low-dose cyclophosphamide to patients with breast cancer decreases the number of circulating Tregs while increasing the number of tumor-specific T cells [23]. This response correlated with improved clinical outcomes in patients with advanced breast cancer [23]. However, in patients with breast cancer who have received NAC, the number of MDSCs increased with doxorubicin and cyclophosphamide administration, suggesting that cyclophosphamide is responsible for the increase in MDSCs [24]. These findings reflect the paradox that cyclophosphamide induces specific suppressor cells that inhibit the immune response while having marked immunostimulatory effects [25].

### 3.4. Methotrexate

Methotrexate inhibits dihydrofolate reductase, which reduces the amount of folate needed for DNA synthesis in cancer cells. It has been found to maintain the ability of 6-sulfo LacNAc(+) DCs, a major DC subpopulation in human blood, to induce proinflammatory cytokines and activate T lymphocytes and NK cells; thus, to exert immunostimulatory effects [26]. However, doxorubicin reduced these abilities [26].

The CD8 T cell-mediated type 1 immune response (Tc1) is associated with cytolytic effects, IFN-γ production, and an effective antitumor response. A single dose of methotrexate prior to Tc1 effector-cell transfer was found to enhance the type 1 antitumor response mediated by CD8 in a transgenic mouse model of T cell receptor breast cancer [27]. Furthermore, the combination of methotrexate and Tc1 effector cells not only allowed donor Tc1 cells to accumulate and persist in the primary tumor growth site, but also increased the donor TIL level [27]. These effects remarkably enhanced the appearance of endogenous differentiated CD8 cells infiltrating the tumor and reduced early Tregs recruitment compared with that in a group that received only methotrexate or Tc1 cell transfer [27]. These results suggest that the combination of Tc1 cell transfer and methotrexate can reduce the number of Tregs in the tumor microenvironment and enhance the antitumor effect [27].

### 3.5. 5-Fluorouracil

5-fluorouracil (5-FU) is a pyrimidine antimetabolite that inhibits the biochemical pathways involved in DNA synthesis; it also inhibits RNA synthesis. Immunomodulatory effects of 5-FU on gastrointestinal tumor cells, such as gastric and colorectal cancer cells, have been observed. Standard doses of 5-FU are thought to exert an immunostimulatory effect by promoting antigen uptake by DCs and sensitizing tumor cells to cytotoxicity by NK and CD8+ T cells [28]. 5-FU selectively eliminated MDSCs, which accumulate with tumor progression and reduce antitumor immune responses in mice with EL-4 thymomas [29]. It may be able to alleviate tumor-induced immunosuppression and restore anticancer immune responses because of its ability to directly kill tumor cells, promote the activation of immune effector cells, and eliminate immunosuppressive MDSCs [47]. On the other hand, 5-FU induced the surface expression of programmed death ligand 1 (PD-L1) in human breast cancer cells and subsequently promoted T cell apoptosis via PD-L1, suggesting a link between chemotherapy and resistance to immunotherapy [48].

### 3.6. Capecitabine

Capecitabine, a nucleoside analog used in patients with metastatic breast cancer, is a prodrug of 5′-deoxy-5-fluorouridine (5′-DFUR), which is converted to 5-FU in the following process: metabolism to 5′-deoxy-5-fluorocytidine by carboxylesterase, conversion to 5′-DFUR by cytidine deaminase, and conversion to 5-FU by thymidine phosphorylase, which is highly expressed in breast cancer [49,50]. Capecitabine induces the death of tumor cells and release of TAs, making tumors more susceptible to detection by the immune system. In this process, antigen-presenting cells are activated and, subsequently, TAs are presented to T cells [30]. Capecitabine may promote a more pronounced immune response in patients with immunogenic tumors, such as TN breast cancer, by preferentially producing high concentrations of fluorouracil in tumor tissue. Capecitabine depleted MDSCs and alleviated their suppressive effects on T and NK cells in tumors, resulting in a net increase in immune activity, including high cytolytic activity and IFN-γ production, which activated components of the key downstream STAT pathway and led to the generation of a more potent antitumor immune response [31].

### 3.7. Trastuzumab

Trastuzumab is a HER-2-targeting antibody. It inhibits signals transmitted by HER-2 and tumor cells’ internalization and degradation of HER-2 and subsequent signaling pathways [36]. Trastuzumab enhances the cytotoxic activity of human HER-2-specific CD8+ CTLs, which is thought to be due to an increase in the number of HER-2 peptides presented in major histocompatibility complex (MHC) class I as a result of HER-2 degradation by the antibody [36]. Trastuzumab treatment was associated significantly with an increase in tumor-associated NK cells and lymphocytes expressing granzyme B and TiA1, supporting a role for the immune response in the antibody mechanisms of action in patients with breast cancer receiving NAC [37]. NK cells kill HER-2-overexpressing breast cancer cells coated with trastuzumab via an antibody-dependent cellular cytotoxicity (ADCC) mechanism mediated by the FcγRIII receptor (CD16) [38].

### 3.8. Pertuzumab

Pertuzumab is a monoclonal antibody against the extracellular dimerization domain of HER-2. Its epitope on HER-2 is distinct from that of trastuzumab, and its binding inhibits the dimerization of HER-2 with other HER family receptors, such as HER-1, HER-3, and HER-4. ADCC activation is the primary mechanism of action of trastuzumab, but pertuzumab can also increase ADCC [39]. Compared with monotherapy, combination therapy with clinically acceptable doses of trastuzumab and pertuzumab was found to promote the recruitment of NK cells that produce ADCC and delay the growth of xenograft tumors from inherently trastuzumab-resistant breast cancer cells [40]. In addition, although trastuzumab exerts only modest complement-mediated cytotoxicity (CDC), a study of the CDC effect of the combination of trastuzumab and pertuzumab on HER-2-positive breast cancer cells showed that the suppression of complement regulatory factor expression by small interfering RNA increases the antitumor activity of trastuzumab and pertuzumab via complements and macrophages [41]. The increased antitumor activity of this combination therapy may be due to the complementarity of trastuzumab and pertuzumab, which inhibit HER-2 dimerization and prevent the formation of p95HER2 [39].

### 3.9. Trastuzumab Emtansine

Trastuzumab emtansine (T-DM1) is an antibody-drug conjugate that combines trastuzumab with the cytotoxic agent DM1. After binding to HER-2, T-DM1 is taken up by endosomes and degraded, releasing DM1. T-DM1 blocks the HER-2 signaling pathway and mediates ADCC. Ansamitocin P3, a precursor of T-DM1, induces DC maturation. It increases the production of maturation markers and proinflammatory cytokines, which in turn promotes antigen uptake, the homing of DCs in tumors to TDLNs, and the activation of tumor-specific T cells [42]. T-DM1 induces DC maturation and stimulates antitumor immunity [43]. In a mouse model, despite inherent resistance to immunotherapy with anti-CTL-associated antigen 4/programmed death 1 (PD-1) agents, the combination of T-DM1 and these agents induced innate and adaptive immunity, resulting in a cure with the development of immunogenic memory [43]. Tumor rejection was accompanied by T cell infiltration, polarization of T helper 1 cells, and a marked increase in Tregs. The decreased Treg concentration resulted in inflammation and tissue damage, suggesting that Tregs play a major role in host protection during treatment [43].

## 4. Molecular Characteristics of Anticancer Agent-Induced ICD

The initial phenomenon of doxorubicin-induced ICD has been characterized using various cancer cells, including colon cancer and melanoma cells, via immunogenic apoptosis [51]. Doxorubicin-induced tumor cell death triggers an effective antitumor immune response, inhibiting inoculated tumor growth or regressing the established cancer. In several models, immunization with doxorubicin-treated tumor cells protected mice from subsequent tumor implantation [51]. Treatment with anthracyclines, such as doxorubicin and epirubicin, and alkylating agents, such as cyclophosphamide, induces ICD, characterized by three essential elements: (i) the exposure of CRT, a Ca^2+^-binding endoplasmic reticulum (ER) chaperone protein, on the membrane surfaces of dying cells and DC promotion of TA uptake [52]; (ii) the release of HMGB1 to facilitate antigen processing and presentation to T cells [53]; and (iii) the secretion of ATP, which activates inflammasomes and proinflammatory cytokine production [54]. During ICD, CRT exposure activates signaling pathways mediated by pro-apoptotic proteins, such as caspase-8 and the Bcl-2 family members (e.g., Bax and Bak), beginning with ER stress and involving soluble N-ethylmaleimide-sensitive factor attachment protein receptor (SNARE)-dependent exocytosis [55]. CRT migrates from the ER around the nucleus to the cell periphery and relocalizes the ER chaperone ERp57 to the plasma membrane, even in immunocompetent dead cells. Exposure of the CRT/ERp57 complex on the cell surface promotes DC phagocytosis [56], leading to TA presentation and tumor-specific CTL responses [52]. Such exposure on the surfaces of dying tumor cells in response to some immunogenic anticancer agents triggers an antitumor immune response before any sign of apoptosis appears. It activates the ER stress-associated PKR-like ER kinase, leading to phosphorylation of the eukaryotic initiation factor 2α (eIF2α). Subsequently, caspase-8 is partially activated, leading to cleavage of the ER protein BAP31, which results in the structural activation of Bax and Bak. CRT that passes through the Golgi apparatus is secreted by exocytosis dependent on SNARE [55].

DAMPs released during ICD bind to phagocytosis receptors in immature DCs with the release of HSP70/90 chaperones: CD91 for CRT, P2Y2R/P2X7R for ATP, TLR4 for HMGB1, and formyl peptide receptor 1 (FPR1) for annexin A1 [57]. Most DAMPs are recognized by PRRs. DAMPs act as a “danger” signal that triggers an immunostimulatory response, including the mobilization and activation of immune cells such as neutrophils and macrophages [58]. CRT-bound TAs and TAAs are engulfed by immature DCs in the presence of proinflammatory cytokines, such as IL-1β, IL-6, and TNF-α, to promote DC maturation. Activated DCs present TAs and TAAs as MHC class I molecules to CD8+ T cells, resulting in the production of CTLs in TDLNs, which are tracked to the tumor site and eradicate tumor cells [4] (Figure 2). ATP acts as a “find me” signal, CRT acts as an “eat me” signal, and CD47 in tumor cells acts as a “do not eat me” signal [56]. The balancing of these signals in tumor cells may be critical for engulfment by immature DCs for immune activation after anticancer agent treatment. The release of DAMPs for immune activation differs depending on the anticancer agent. Membrane-exposed CRT has been observed with anthracyclines, cyclophosphamide, and taxanes, whereas ATP secretion has been observed only with anthracyclines and HMGB1 release was not observed with docetaxel; the latter was determined to represent immunogenic modulation [59]. The effect of such differences on the magnitude of immune activation to eradicate breast cancer cells remains unclear. What is known is that the induction of ICD involves classical ICD, non-classical ICD, and immunogenic modulation leading to immune activation against tumor cells.

FPR1 is a PRR that not only recognizes bacterial N-formylated peptides, but also interacts with several DAMPs, including annexin A1. The administration of cyclosporine H, an FPR1 antagonist, abolishes the antitumor effect of chemotherapy, suggesting that FPR1 expression in the host and annexin A1 expression in tumor cells are required for the chemotherapeutic inhibition of tumor growth. FPR1 was not required for DCs to mobilize into tumors, but it was necessary for DCs to approach dying tumor cells, take up TAAs, and cross-present to T cells [60]. Single nucleotide polymorphisms that suppress FPR1 signaling are associated with decreased survival in patients receiving adjuvant chemotherapy for breast cancer, suggesting that FPR1 is important for the antitumor immune response to chemotherapy [60].

## 5. Immunogenic Modulation by Anticancer Agents as Non-Classical ICD

The molecular cascade leading to the release of HMGB1 has not been identified definitively but is thought to occur through a caspase-dependent mechanism [54]. Cellular mechanisms of autophagy [61] and pannexin-1 channels [62] also may be involved in ATP secretion [63]. The failure of cell death via the intrinsic processes of CRT exposure, HMGB1 release, and ATP secretion is nonimmunogenic. ICD was recently shown to occur with immunogenic modulation by docetaxel; this drug modulates the antitumor immune response but does not induce classical ICD [59]. In a breast cancer model, docetaxel administration did not induce HMGB1 or ATP secretion or cell death, and tumor cells were not killed by ICD. However, in breast cancer cell lines examined after docetaxel treatment, exposure to CRT was observed, and the killing ability of CTLs was enhanced by the increased components of the antigen-processing system and membrane translocation of CRT [59]. Furthermore, docetaxel-resistant cells are not resistant to killing by CTLs, suggesting that tumor cell exposure to nonlethal or sublethal doses of docetaxel changes the phenotype of the tumor, making it more sensitive to killing by CTLs [59]. In similar lines, treatment with paclitaxel or cyclophosphamide did not result in ATP secretion in breast cancer cells. DAMP responses induced by anticancer drugs may depend on the cancer type and drug dose [64].

## 6. Roles of DAMPs and ICD in the Clinical Significance of Breast Cancer

HMGB1 is a ubiquitous non-histone chromosomal protein that is enriched in activated chromatin and plays tumor-promoting and antitumor roles in tumorigenesis. The elucidation of its structure and function may inform the development of new breast cancer therapies [e.g., new drugs, miRNA (miR) utilization, cancer stem cell (CSC) targeting, and immunotherapy] [65].

The dynamics of HMGB1 in serum during NAC correlate with the treatment response in patients with advanced breast cancer, suggesting that the serum HMGB1 level is a circulating biomarker of ICD and treatment response [66]. Furthermore, among patients with early-stage breast cancer who undergo NAC, those with increased HMGB1 levels had significantly better overall survival (OS) than did those with smaller HMGB1 changes, suggesting that immediate HMGB1 level augmentation is associated with better prognosis in these patients [67]. However, although another group of patients with breast cancer undergoing NAC had increased levels of CRT as well as HMGB1, neither increase correlated significantly with pathological effects or survival [68]. HMGB1 secretion promotes fibroblast activation via receptor for advanced glycation end products (RAGE)/aerobic glycolysis, and activated fibroblasts promote breast cancer metastasis via an increase in lactate [69]. HMGB1 is expressed highly in patients with TN breast cancer and may be involved in lung metastasis in combination with CD62L^dim^ neutrophils [70]. It also has been shown to promote tumor angiogenesis via hypoxia inducible factor (HIF)-1α, which simulates the PI3K/AKT signaling pathway in breast cancer cells [71]. Thus, HMBG1 secretion may be involved in tumor progression and distant metastasis, depending on the magnitude and duration of its release in a proinflammatory response to external stimuli; it may be beneficial or detrimental to immune activation and tumorigenesis [72]. Tumorigenesis and breast cancer progression induced by HMGB1 as a ligand may be mediated via the TLR4/myeloid differentiation factor 88 and RAGE signaling pathways for tumor invasion and metastasis, CSC renewal, epithelial–mesenchymal transition (EMT), drug resistance, and cancer recurrence [73,74]. As for the potential role of CSC maintenance in tumor recurrence and metastasis, autophagic cancer-associated fibroblasts (CAFs) have been shown to be important for luminal breast cancer progression promotion via the HMGB1-TLR4 signaling pathway, and CAF autophagy and TLR4 in breast cancer cells are potential therapeutic targets [75]. Several miRs exert antitumor effects by targeting HMBG1 in breast cancer cells: miR-107 has been shown to inhibit the autophagy, growth, and migration of these cells in vitro and in vivo [76], and miR-205 has been shown to decrease the viability and acquisition of TN breast cancer cells in the EMT phenotype by partially targeting the HMGB1-RAGE signaling pathway [77].

The overexpression of HMGB1, a key target of miR-142-3p in breast cancer cells, restored doxorubicin resistance by promoting apoptosis and inhibiting autophagy via miR-142-3p upregulation, suggesting that the miR-142-3p/HMBG1 axis is a novel target to overcome drug resistance in patients with breast cancer [78]. In contrast, another study showed that miR-129-5p enhances the sensitivity of breast cancer cells to paclitaxel by inhibiting autophagy and HMGB1, thereby increasing apoptosis [79]. In addition, patients with endocrine therapy (ET)-refractory HR-positive breast cancer showed greater HMGB1 expression correlated with worse postoperative progression-free survival (PFS) [80]. Among patients with ET-resistant disease, treatment with cyclin-dependent kinase 4 and 6 (CDK4/6) inhibitors benefitted the PFS of HMGB1-positive patients relative to that of HMGB1-negative patients [80]. HMGB1 promotes tamoxifen resistance via TLR4, resulting in the activation of the NF-κB signaling pathway, which is restored by CDK4/6 inhibitors [80]. These findings suggest that HMGB1 is involved in the promotion of the ET resistance of HR-positive breast cancer. HMGB1 was found to be the primary autophagy target of the XIAOPI formula, a new anti-mammary hyperplasia drug that inhibits breast cancer metastasis [81]. This formula increased the paclitaxel chemosensitivity of breast cancer cells via the chemokine (C-X-C motif) ligand 1/HMGB1 autophagy axis [81]. Similarly, the suppression of mediator complex subunit 19 down-regulated autophagy by inhibiting HMGB1 signaling and enhanced doxorubicin chemosensitivity in human breast cancer cells [82]. Thus, HMGB1 is involved in drug resistance via the positive regulation of autophagy in breast cancer cells, as in other types of cancer cell [83]. ER stress induces HMGB1 secretion in TN breast cancer cell lines, and the cytoplasmic expression of HMGB1 correlates positively with the abundance of TILs in TN breast cancer, suggesting that HMGB1 secretion mediates subsequent tumor-specific immune activation by CTLs [84]. Similarly, greater cytoplasmic HMGB1 expression has been reported in TN and HER-2-positive breast cancers than in HR-positive tumors; in TN breast cancer, this expression, but not nuclear HMGB1 expression, was associated significantly with TIL and CD8+ cell abundance [85].

Patients with breast cancer and hypofunctional alleles of TLR4, TLR3, and FPR1, which interact with HMGB1, double-stranded RNA, and annexin A1, respectively, had lower rates of OS and metastasis-free survival than did their normal-allele counterparts after anthracycline-based adjuvant chemotherapy, suggesting that conventional adjuvant chemotherapy activates antitumor immunity against TAAs from dead cells via PRRs [86]. The immune response to anticancer drugs determines the success of this type of adjuvant chemotherapy, due partly to the ability of anthracyclines to induce ICD with HMGB1 release prior to autophagy. The lack of autophagy and reduced HMGB1 expression have been associated with extremely poor prognosis in breast cancer specimens [87] and potentially with adverse effects on immune surveillance by anticancer drugs, and thereby with the promotion of tumor progression [88].

CRT is overexpressed in invasive breast cancer tissues relative to normal tissues, suggesting that it is involved in this cancer phenotype through dysregulation of the p53 and mitogen-activated protein kinase signaling pathways [89]. CRT overexpression in breast cancer specimens also may be involved in invasive breast cancer progression, rather than the development of an immune response in breast cancer cells [90]. Similarly, although CRT expression was shown to correlate with estrogen receptor status, CRT was found to independently affect tumor size and distant metastasis, suggesting that CRT expression is associated with more-advanced tumors and is a potential prognostic factor for patients with breast cancer [91]. CRT can act as an effective immunological adjuvant, as its translocation with TAAs to the cell surface induces a tumor-specific immune response via subsequent CRT-mediated signals [92]. Apoptosis was found to be more likely to occur in human breast cancer cells transfected with CRT than in control cells, suggesting that the upregulation of specific proteins in the ER lumen triggers cell death [93].

In overcoming P-glycoprotein (P-gp)-expressing doxorubicin resistance in breast cancer cells, the clinically approved P-gp inhibitor tariquidar and its derivatives restored doxorubicin accumulation and enhanced ICD, defined as an increase in CRT translocation and the release of HMGB1 and ATP [94]. Thus, pharmacological compounds that abolish drug resistance may act as chemosensitizers and immunosensitizers, increasing the therapeutic effect. Doxorubicin was found to increase the level of indoleamine 2,3-dioxygenase (IDO), an enzyme that converts tryptophan to kynurenines and is involved in the suppression of the immune response of breast cancer cells in murine models, but in combination with a small molecule of an IDO inhibitor (NLG919) it enhanced ICD, inducing a synergistic therapeutic effect via the reversal of CD8+ T cell suppression [95]. The combination of the anti-CD47 antibody and anthracyclines has shown good therapeutic efficacy for orthotopic breast cancer; this antibody eliminates M2 macrophages and Tregs in the tumor microenvironment [96].

The extracellular ATP concentration in the tumor microenvironment increases with cell death caused by chemotherapeutic agents, such as doxorubicin, and the interaction of extracellular ATP with the purine receptor P2X7 causes breast cancer cell migration and invasion by increasing cyclooxygenase 2 expression [97]. The P2X7 receptor is an ATP-gated, nonselective cation channel receptor involved in signal transduction, cytokinesis, and tumor growth regulation and cell development. Increased P2X7 receptor expression promotes breast cancer invasion and migration via the AKT signaling pathway and EMT, which are influenced by factors such as hypoxia and ATP expression [98]. miRs and P2X7 receptor inhibitors can inhibit P2X7 receptor-mediated breast cancer development and invasion [98]. In addition, extracellular ATP promotes the invasion and chemoresistance of breast cancer cells through the sex-determining region Y-box 9 (SOX9) signaling pathway, which interacts with IL-6–Janus kinase 1–STAT3 signaling [99]. The treatment of SOX9-knockdown breast cancer cells with apyrase, an ATP hydroxylase, suppressed tumor growth and increased drug sensitivity in nude mice [99]. The upregulation of ATP–SOX9 signaling was associated with poor prognosis in patients with breast cancer [99]. In another study, extracellular ATP promoted breast cancer invasion and EMT via ATP–HIF-2α signaling, targeting ATP-driven invasion via matrix metalloproteinase-9 and ATP-driven EMT via E-cadherin and Snail, respectively [100]. Molecules involved in this signaling are expressed highly in human breast cancer tissue and are associated with poor prognoses [100]. Breast cancer cells with high metastatic potential, such as MDA-MB-231 cells, are characterized by greater ATP release and P2Y2 receptor (P2Y2R) activity relative to less-metastatic cells. The ATP-mediated activation of P2Y2R plays important roles in tumor progression and metastasis. P2Y2R expression correlates significantly with Notch-4 alone, a CSC marker, in patients with breast cancer [100,101]. The P2Y2R and P2X7 receptor may engage in crosstalk or cooperation in the processes of immune response and tumor growth [101]. Increased ATP synthesis is also involved in the acquired drug resistance of breast cancer cells to HER-2-targeted antibody therapy, and such resistance can be reversed via the inhibition of ATP synthesis by oligomycin A or knockdown of the ATP synthase complex in resistant cells [102]. The overexpression of ATP-binding cassette transporter proteins, such as P-gp, multidrug resistance-associated proteins, and ATP-dependent breast cancer resistance proteins, may be involved in ATP-mediated multidrug resistance [103]. Thus, intracellular and extracellular ATP may be involved in tumor progression, metastasis, drug resistance, and immune activation as DAMPs that depend on external stimuli and their magnitudes in the breast tumor microenvironment. The dual roles of DAMPs in antitumor immunity and tumor development in the context of breast cancer are summarized in Figure 3.

## 7. Exploiting the Double-Edged Sword of DAMPs for Antitumor Immunity

DAMPs may have a cytoprotective function by triggering host defense immunity through PRRs that detect dying tumor cells and by promoting the ICD of these cells. During chemotherapy-induced ICD of tumor cells, large numbers of PRR ligands and DAMPs, such as DNA and RNA, are released to further activate PRRs and enhance antitumor immunity. DAMPs serve as ligands for TLRs expressed on immune cells, inducing cytokine production and T cell activation. Cell death and DAMP release may also lead to chronic inflammation, thereby promoting tumor progression. Tumor cells secrete DAMPs for the differentiation of naive T cells into Tregs and secretion of cytokines such as IL-10 and TGF-β to promote the formation of an immunosuppressive microenvironment. Chronic inflammation in the tumor microenvironment induced by DAMPs causes increases in the numbers of immunosuppressive immune cells, such as tumor-associated macrophages, MDSCs, and Tregs. In addition, tumor cell-derived DAMPs directly activate TLRs in tumors, inducing chemotherapy resistance, migration, invasion, and metastasis. Thus, DAMPs play dual roles in tumor growth and antitumor immunity [104]. How efficiently DAMPs are involved in the induction of antitumor immune responses in the tumor microenvironment is not clear, but the induction of ICD and immunogenic effects by anticancer agents may induce antitumor immunity in patients with breast cancer via DAMP release under certain circumstances. Further studies conducted with endogenous and anticancer drug-treated tumor models are needed to clarify the relevance of ICD and DAMPs in the activation of host immune defenses.

Gemcitabine, used to treat breast cancer recurrence, induces the release of characteristic immunostimulatory DAMPs, such as HMGB1, CRT, and HSP70, in bladder cancer cells, but does not cause ICD of these cells [105]. The release of prostaglandin E_2_ inhibits the immunostimulatory effects of DAMPs, and prostaglandin E_2_ blockade activates DCs, followed by the priming of antitumor immune responses by CD8+ T cells, leading to tumor rejection and the failure to induce ICD [105]. These findings suggest that the balance between immunostimulatory and inhibitory DAMPs determines the induction of ICD by anticancer agents. Whether they are applicable to breast cancer cells remains unclear.

## 8. Immunomodulatory Effects of Anticancer Agents and the Enhancement of Antitumor Immunity in Combination with Immune Checkpoint Inhibitors

Breast cancer has been thought to be a “cold” tumor, with low immunogenicity and a relatively low mutation burden. Nevertheless, high TIL and PD-L1 expression has been observed recently in TN and HER-2-positive breast cancers and some luminal breast cancers [106]. Immune checkpoint inhibitors (ICIs) that target PD-1 and PD-L1 improve therapeutic efficacy by enhancing immunogenicity and modulating the tumor microenvironment. Anticancer drugs, such as anthracyclines and taxanes, increase PD-L1 expression in tumor cells. Thus, combination therapies with anticancer drugs and ICIs enhance the therapeutic effect in patients with breast cancer [107]. Pivotal trials examining metastatic TN and HER-2-positive breast cancer, including the phase-III KEYNOTE-119 [108], IMpassion130 [109], KEYNOTE-355 [110], phase-II KEYNOTE-086 [111], and phase-Ib/II PANACEA [112] trials, have revealed that the good OS, PFS, and response rates achieved with ICI use may be associated with PD-L1 positivity and TIL abundance in tumors [113], and that TILs are surrogate markers for existing antitumor immunity. Regarding the efficacy of ICI use in combination with anticancer drugs, the phase-II TONIC study suggested that the use of nivolumab, an anti-PD-1 antibody administered after induction with doxorubicin, resulted in a higher response rate than did the use of cisplatin or cyclophosphamide in patients with metastatic TN breast cancer [114].

The KEYNOTE-173 trial was a six-cohort phase-Ib study in which taxanes were administered with or without carboplatin, followed by doxorubicin/cyclophosphamide (AC) administration in combination with pembrolizumab at different doses and schedules and with a phase-II dose recommended, in the NAC setting for TN breast cancer [115]. pCR rates ranged from 30% to 80%, and higher pCR rates were associated with greater PD-L1 positivity and stromal TIL abundance before treatment [115]. In the phase-II I-SPY2 trial, patients were randomized to receive pembrolizumab in combination with paclitaxel followed by AC [116]. The combination treatment with pembrolizumab more than doubled pCR rates in patients with TN, HER-2-negative, and HR-positive/HER-2-negative breast cancers. TIL infiltration and the density of Tregs and PD-1-positive T cells correlated with pCR [116]. The macrophage to Tc cell ratio inversely predicted resistance to pembrolizumab in combination with anticancer agents [117]. In the phase-II GeparNuevo trial, patients with TN breast cancer were randomized to receive durvalumab, anti-PD-L1 antibody, or placebo followed by nab-paclitaxel and epirubicin/cyclophosphamide [118]. The pCR rate was greater in patients treated with durvalumab before NAC than in those who received chemotherapy alone, suggesting that such priming with durvalumab is important for the induction of an antitumor immune response. The increase in pCR was associated with an increase in interstitial TILs in both groups. PD-L1 positivity in tumor cell components and immune cells predicted the response to treatment with durvalumab and anticancer agents.

The double-blind phase-III KEYNOTE-522 trial compared pembrolizumab/NAC followed by adjuvant pembrolizumab monotherapy with neoadjuvant paclitaxel/carboplatin and AC or epirubicin/cyclophosphamide plus placebo therapy followed by adjuvant placebo for TN breast cancer [119]. The pCR rate was significantly higher for pembrolizumab/NAC than for NAC (64.8% vs. 51.2%), but the increase in pCR achieved with pembrolizumab was not associated with PD-L1 positivity, as it was in the KEYNOTE-173 trial [119]. The reason for this difference is not clear, but the dosing schedule used in the KEYNOTE-522 trial may have induced more PD-L1 expression than did that used in the KEYNOTE-173 trial. The magnitude of pathologic response in the KEYNOTE-522 trial may be due to the contribution of anthracycline chemotherapy, which may have a preferential immunostimulatory effect [120]. Furthermore, in the phase-III IMpassion031 trial, atezolizumab and nab-paclitaxel administration followed by AC administration resulted in a higher pCR rate than did placebo/chemotherapy, regardless of PD-L1 expression [121].

Several phase-III trials of (neo)adjuvant therapy with ICIs are underway, with expansion of this treatment approach to reduce recurrence in high-risk patient populations (i.e., patients with TN, HR-positive/HER-2-negative, and HER-2-positive breast cancers; lymph node metastasis; or locally advanced cancer with residual tumor cells after NAC). In these studies, ICIs are intercalated into established (neo)adjuvant chemotherapy or sequential treatment after NAC or surgical treatment. Although further studies are needed to clarify the ability of such combination therapies to induce ICD, activate antitumor immune responses, and elicit long-term antitumor immune memory for the curing of primary breast cancer, the use of ICIs in combination with anticancer agents in the (neo)adjuvant setting appears to be a promising strategy for the modulation of the immunosuppressive microenvironment and enhancement of the antitumor effect, ultimately for a cure.

## 9. ICD and Long-Term Antitumor Immune Memory

The mechanisms of ICD induction have been classified into two types [4]. Type I ICD induces ICD-related danger signals as a result of the primary effects of ER stress, depending on the site of off-target or secondary effects. Type II ICD, such as that induced by hypericin-photodynamic therapy, selectively targets the ER and induces cell death and danger signals causing ICD-associated immunogenicity. Anticancer agent-induced ICD is type I ICD, related closely to the unfolded protein response (UPR) ER stress pathway, which is involved in the induction of autophagy. For example, anthracyclines exert their cytotoxic effects primarily in the localized nucleus, with ER stress occurring as a secondary effect. This type of ICD may induce an anticancer vaccine effect, i.e., the generation of an in situ vaccine for long-term anticancer agent immunity. Nanopulse stimulation-induced ICD, achieved by the ablation of orthotopic 4T1 mouse breast cancer, resulted in the activation of DCs by cell death and increased the number of memory effector T cells in the blood, exerting an abscopal effect that prevented distant metastasis [122]. The killing of cancer cells in a way that generates long-term antitumor immunity similar to the natural antitumor immune response, based on the induction of ICD, is essential for a cancer cure.

In patients with breast cancer receiving adjuvant chemotherapy or NAC, whether anticancer agent-induced ICD results in the production of memory effector T cells to prevent distant recurrence is unclear. However, the acquisition of immune privilege via anticancer agents may be involved in the long-term antitumor immunity that could lead to a primary breast cancer cure (Figure 4) [123]. When antitumor immunity is not acquired successfully, drug-resistant or dormant cells that form and persist in niches in distant organs and in the blood can promote tumor growth and lead to distant metastasis. During tumor progression, tumor cells must reorganize their microenvironment by inducing ER stress, such as via hypoxia, an acidic pH, and nutrient deprivation [124]. Hypoxia is known to be an important factor causing drug resistance and immune evasion. It induces ER stress leading to ICD via CRT exposure in human and mouse breast cancer cell lines, suggesting that it enhances the immunogenicity of breast cancer cells in the immunosuppressive tumor microenvironment [125]. UPR activation is an important feature of tumor development, associated with the ability to acquire essential features such as tumor dormancy, drug resistance, and tumor angiogenesis [126]. However, the relationship among ER stress, UPR activation, and the acquisition of these tumor characteristics after chemotherapy in patients with primary breast cancer remains to be elucidated.

## 10. Future Therapies

IMMUNEPOTENT CRP (ICRP) is a mixture of low-molecular-weight substances obtained from bovine spleen that has immunomodulatory and cytotoxic effects on cancer cells, produced by the Laboratory of Immunology and Virology of the Autonomous University of Nuevo León, Mexico [127]. The combined administration of chemotherapy and ICRP improves clinical parameters in patients with breast cancer. In a mouse model of TN breast cancer, treatment with AC and ICRP reduced tumor volume, prolonged survival, increased the numbers of infiltrating and systemic CD8+ T cells, and decreased the numbers of tumor suppressor molecules such as galectin-3, PD-L1, and IL-10 [128]. ICRP regulated the tumor microenvironment and immune response, enhancing or prolonging the antitumor effects of AC [128]. In a preclinical antitumor immune-memory model, ICRP induced caspase-independent reactive oxygen species-dependent cell death, eIF2α phosphorylation, autophagosome formation, and the release of HMBG1 and ATP in breast cancer cells [129]. ICRP-induced cell death promoted the maturation of DCs, triggering T cell priming and CTL-induced breast cancer cell death [129]. Prophylactic vaccination against ICRP-induced cell death prevented tumorigenesis in BALB/c mice; induced long-term antitumor memory with DC maturation in lymph nodes; increased the numbers of CD8+ T cells in the lymph nodes, peripheral blood, and tumor sites; and showed ex vivo tumor-specific cytotoxicity by splenocytes [129]. Thus, ICRP-induced ICD leads to long-term antitumor memory in breast cancer cells.

Breast cancer vaccines may be another way to gain long-term antitumor immunity and cure breast cancer. Although breast cancer cells are created by genetic mutations from normal epithelial cells, the immunosuppressive network in a breast tumor leads to the evasion of immune cell attack and non-recognition of T cells by immune cells [130]. The identification of specific genetic mutations, such as those in TAs, TAAs, and neoantigens detected by methods such as next-generation sequencing, enables the development of a unique vaccine that is recognized by DCs as foreign and activates tumor-specific T cells, eradicating residual tumor cells that are not detected after surgical treatment and inhibiting breast cancer recurrence. The administration of a vaccine combined with immune-adjuvant GM-CSF or irradiated tumor cells can enhance the vaccine-derived immune response. Furthermore, the combined use of anticancer agents, including ICIs, and vaccines can enhance the antitumor immunity provided by the vaccines by utilizing the ICD and immunogenic modulation caused by the anticancer agents. The antitumor efficacy of peptide-based, whole-tumor, gene-based, DC-based, and RNA vaccines for breast cancer is being evaluated in preclinical studies and clinical trials [131,132]. Nanotechnology appears to be a promising means of developing effective immunotherapeutic vaccines for breast cancer [131]. In particular, vaccines targeting HER-2 have been developed in preclinical and clinical studies, and may be promising for the enhancement of the antitumor immune response through the provision of CTLs and memory CTLs in combination with HER-2-targeting agents for different HER-2 expression levels in patients with breast cancer [133]. Further studies are required to establish vaccine therapies that overcome the shortcomings of conventional therapies and induce long-term antitumor protective immunity leading to a cure for breast cancer.

Finally, to overcome the issue of tumor heterogeneity in breast cancer treatment, a recent study involved the application of a network-based approach to the Cancer Genome Atlas data stratified by clinical and genetic classifications [134]. Four switch proteins that are overexpressed in all tumor subtypes were identified [134]. The inhibition of aurora kinase A (AURKA), one of the four switch genes identified, inhibited tumor growth via apoptosis similarly in all tumor subtypes in in vitro and in vivo experiments [134]. A recent randomized clinical trial showed that the weekly administration of paclitaxel with alisertib based on elevated AURKA expression significantly improved the PFS of patients with metastatic HR-positive and HER-2-negative breast cancer compared with the use of paclitaxel alone; the efficacy of this treatment in patients with TN breast cancer was not examined due to late enrollment [135]. Alisertib has been shown to alter the immunosuppressive state by eliminating MDSCs via apoptosis and to enhance antitumor immunity by increasing CD8+ and CD4+ T cell abundance, leading to the regression of mouse mammary tumors [136]. Furthermore, combination therapy with alisertib and anti-PD-L1 antibody synergistically enhanced the antitumor effect in mouse mammary tumors [136]. Further studies of novel chemoimmunotherapies with alisertib and ICIs for advanced breast cancer are needed.

## 11. Concluding Remarks

At present, whether anticancer agent-induced ICD can provide long-term antitumor immune memory and lead to a primary breast cancer cure remains unclear. Although preclinical studies have suggested that ICD activates tumor-specific immune responses by CTLs, which serve as long-term immune memory and inhibit distant recurrence, no available clinical evidence supports the involvement of such antitumor immune memory in the curing of primary breast cancer. As NAC induces antitumor immune activation for specific populations, including patients with luminal, TN, and HER-2-positive breast cancers, the activated tumor-specific immune response may eradicate residual tumor cells and contribute to the curing of breast cancer in these patients. Adjuvant chemotherapy may activate antitumor immune responses for long-term immune memory with small residual tumors, but little DAMP release from ICD. Regardless of whether chemotherapy is neoadjuvant or adjuvant, anticancer agent-induced ICD may trigger the acquisition of antitumor immune memory in individual patients with breast cancer. The immune factors required for permanent control and the acquisition of long-term antitumor immunity in this context need to be identified, and combination therapy with anticancer agents and ICIs or vaccines, and novel anticancer agents under development, may enhance antitumor immunity by modulating the immunosuppressive situation in the tumor microenvironment, leading to the curing of primary breast cancer.

## Figures and Tables

**Figure 1 cancers-13-04756-f001:**
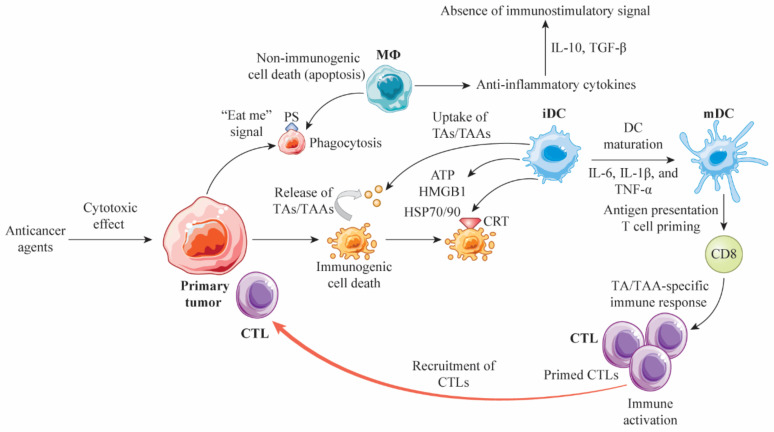
Non-immunogenic and immunogenic cell death caused by anticancer agents. Anticancer agents kill tumor cells via cytotoxic effects. PS externalization in apoptotic cells generates an “eat me” signal, leading to engulfment by macrophages; anti-inflammatory cytokines, such as TGF-β and IL-10, are released, suppressing the immunostimulatory response. In tumor cells dying due to the administration of anticancer drugs, CRTs are exposed on the membrane; DAMPs, such as HMGB1, ATP, and HSP70/90, are released; and pro-inflammatory cytokines, such as IL-1β, IL-6, and TNF-α, are released prior to apoptosis, resulting in the induction of an antitumor immune response known as immunogenic cell death. TAs and TAAs released from dying tumor cells are engulfed by iDCs and then activated into mDCs, which present TAs and TAAs to CD8+ T cells and produce tumor-specific CTLs in tumor-draining lymph nodes. Activated CTLs are recruited into the primary site and lyse the tumor cells. Abbreviations: MФ, macrophage; IL, interleukin; TGF-β, transforming growth factor β; PS, phosphatidyl serine; TA, tumor antigen; TAA, tumor-associated antigen; iDC, immature dendritic cell; DC, dendritic cell; mDC, mature dendritic cell; ATP, adenosine triphosphate; HMGB1, high mobility group box 1; HSP, heat shock protein; TNF-α, tumor necrosis factor-α; CTL, cytotoxic T lymphocyte; CRT, calreticulin; DAMP, damage-associated molecular pattern.

**Figure 2 cancers-13-04756-f002:**
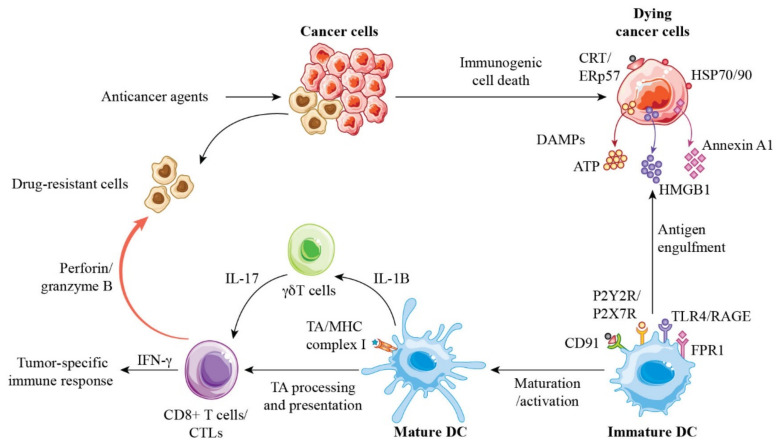
Molecular mechanisms by which immunogenic cell death due to anticancer drugs leads to tumor-specific immune activation. When anticancer drugs are administered, dying tumor cells expose CRT on membrane surfaces in a CRT/ERp57 complex and release DAMPs such as HMBG1, ATP, and annexin A1, which bind to CD91, TLR4/RAGE, P2Y2R/P2X7R, and FPR1, respectively. TAs released from dying cells are engulfed by immature DCs, which are activated and mature. Thereafter, TA and TAA processing proceeds in mature DCs, and TAs and TAAs are presented to CD8+ T cells as MHC complex I molecules, generating CTLs for tumor-specific immune responses. These processes result in IL-1β- and IL-17-dependent and IFN-γ-mediated immune responses involving γδ T cells and CTLs, ultimately leading to the eradication of drug-resistant tumor cells and allowing for the development of long-term immune memory for a cancer cure. Abbreviations: ER, endoplasmic reticulum; RAGE, receptor for advanced glycation end products; FPR1, formyl peptide receptor 1; MHC, major histocompatibility complex.

**Figure 3 cancers-13-04756-f003:**
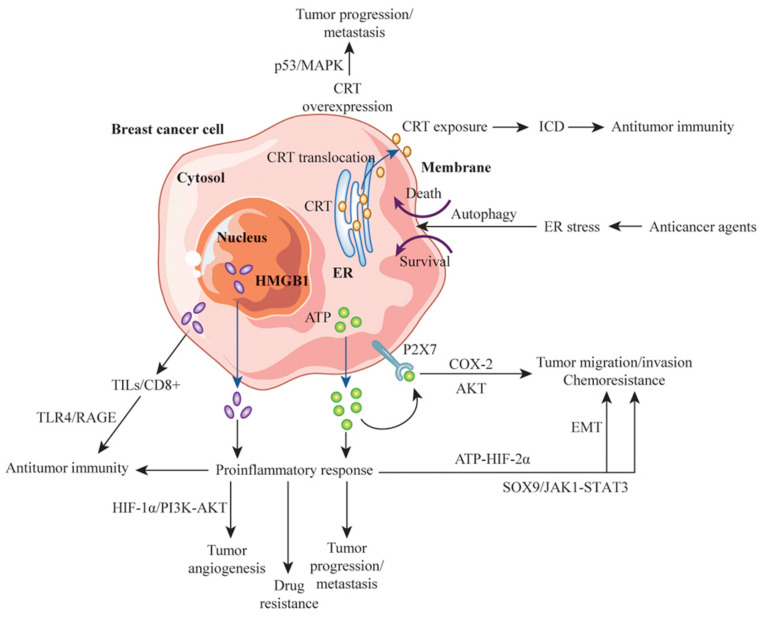
The dual roles of DAMPs in antitumor immunity and tumor development in breast cancer cells. Anticancer agents induce an ER stress response and autophagy, which determines whether cells die or survive. When cells die, CRT in the ER lumen migrates to the tumor cell membrane, exposing the CRT/ERp57 complex as an “eat me” signal. This signal induces an antitumor immune response via ICD-mediated signals. CRT overexpression is associated with tumor progression and metastasis via the MAPK signaling pathway. Nuclear HMBG1 is released extracellularly to trigger inflammatory responses and is involved in tumor progression, metastasis, angiogenesis, drug resistance, and antitumor immunity in response to the tumor microenvironment and external stimuli. In TN breast cancer, cytoplasmic HMGB1 is associated with increased numbers of TILs, which are involved in antitumor immunity. ATP, which is secreted abundantly extracellularly, binds to P2X7 receptors on breast cancer cells and is involved in tumor migration and invasion. Extracellular ATP also triggers an inflammatory response and is involved in tumor migration, invasion, and chemotherapy resistance due to EMT. DAMPs, such as CRT, HMBG1, and ATP, in breast cancer cells play dual roles in tumor progression and antitumor immunity, depending on the magnitude and duration of external stimuli in the tumor microenvironment. Abbreviations: MAPK, mitogen-activated protein kinase; COX-2, cyclooxygenase 2; EMT, epithelial—mesenchymal transition; HIF, hypoxia inducible factor; SOX9, sex-determining region Y-box 9; JAK1, Janus kinase 1; TN, triple negative.

**Figure 4 cancers-13-04756-f004:**
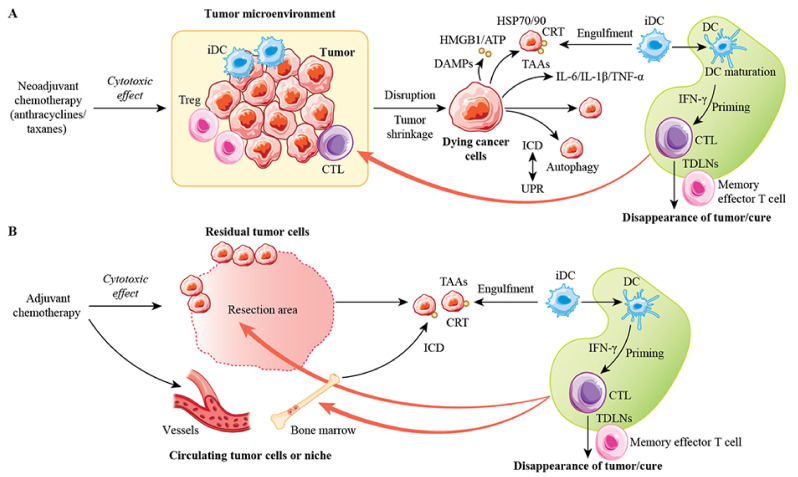
Hypotheses for the means by which immune activation leads after (neo)adjuvant chemotherapy to the curing of primary breast cancer. (**A**) A model of the hypothesis that immune activation after neoadjuvant chemotherapy eradicates tumor cells to cure breast cancer. Anticancer agents such as anthracyclines and taxanes are cytotoxic to tumor cells, disrupt the tumor microenvironment, induce ICD, and cause DAMP processes such as CRT membrane exposure, ATP secretion, and HMGB1 release from dying tumor cells into the extracellular environment. These processes lead to tumor-specific immune activation with the release of pro-inflammatory cytokines such as IL-6, IL-1β, and TNF-α, which alters the immunosuppressive microenvironment and promotes immune activation. Dying cancer cells with exposed CRT containing TAAs are taken up by iDCs, causing DC maturation and CD8+ T cell activation, which in turn generates CTLs in TDLNs. These CTLs may be recruited to the primary tumor site to eradicate tumor cells, which in turn produce memory effector T cells to prevent recurrence. (**B**) A model demonstrating the efficacy of adjuvant chemotherapy in the eradication of residual tumor cells adjacent to surgical resection sites and persistent micrometastases circulating in peripheral blood vessels and distant organ niches. Adjuvant chemotherapy has cytotoxic effects on residual tumor cells and dying cancer cells, with the exposure of CRT containing TAAs that are taken up by iDCs, leading to DC maturation, the induction of CD8+ T cells, and the activation of CTLs in TDLNs. These CTLs may be recruited to the resection site, blood circulation, and distant organs to eradicate remaining tumor cells and reliably treat breast cancer. Regardless of whether chemotherapy is neoadjuvant or adjuvant, tumor-specific immune activation via ICD by anticancer agents can completely eradicate residual tumor cells and contribute to long-term outcomes in patients with primary breast cancer by providing permanent immunity with memory effector T cells to prevent recurrence. Abbreviations: TDLN, tumor-draining lymph node; UPR, unfolded protein response.

**Table 1 cancers-13-04756-t001:** Summary of the mechanistic actions and immunomodulatory effects of anticancer agents used for primary breast cancer treatment.

Anticancer Agents	Mechanism of Action	Immunomodulatory Effect
Anthracyclines		
Doxorubicin	Interferes with the correct unwinding of DNA during replication and transcription	Production of IFN-γ by CTLs, production of IL-17 by γδ T cells and promotion of CTL accumulation by γδ T cells, and IL-1β induction [20,21]. Reduction of MDSCs [22]. Increase in CD4, CD8, and NK cells and expression of IFN-γ, perforin, and granzyme B [22].
Epirubicin	Inhibits the synthesis of nucleic acids and proteins, and forms complexes with DNA by base pair intercalation, thereby inhibiting topoisomerase II activity Promotes ICD	
Alkylating agent		
Cyclophosphamide	Suppresses protein synthesis by inhibiting the transcription of DNA to RNA Promotes ICD	Suppression of Tregs at low dose and Increase in tumor-specific T cells [23]. Increase in MDSCs due to AC as a paradoxical effect [24,25].
Antimetabolites		
Methotrexate	Inhibits dihydrofolate reductase	Maintenance of the ability of DCs to produce pro-inflammatory cytokines and activate NK and T cells [26]. Combination with methotrexate and Tc1 cell transfer increases TILs and decreases Tregs [27].
5-Fluorouracil	Inhibits DNA and RNA synthesis	Promotion of antigen uptake by DCs and enhanced cytotoxicity of NK and CD8+ T cells [28]. Elimination of MDSCs [29].
Capecitabine	Prodrug that converts 5′-deoxy-5-fluorouridine to 5-fluorouracil Inhibits DNA and RNA synthesis	Release of antigens by tumor cell death, activation of DCs, presentation of tumor antigens to T cells [30]. Depletion of MDSCs, alleviation of NK and T cell suppression [31]
Taxanes		
Paclitaxel	Changes tubulin polymerization or depolymerization	Increased levels of IFN-γ, IL-2, IL-6, GM-CSF, and pNK cell activity [32]. Decreased Tregs, independently of TLR4 [33]. Increased permeability of granzyme B by perforin independent CTLs [34].
Nab-paclitaxel	Paclitaxel bound to albumin nanoparticles. Transport of albumin by transcytosis increases its delivery to tumors	
Docetaxel	Promotes immunogenic modulation	Leads MDSCs to transform from the M2 to the M1 phenotype via STAT3 [35].
Targeting agents		
Trastuzumab	Inhibits signals transmitted by HER-2 and degrades HER-2	Enhanced cytotoxicity of HER-2-specific CD8+CTLs [36]. Increase in tumor-associated NK cells and lymphocytes expressing granzyme B and TiA1 [37]. ADCC mediated by FcγRIII receptor [38].
Pertuzumab	Inhibits dimerization between HER-2 and HER family receptors	Activation of ADCC [39]. Mobilization of NK cells by ADCC [40]. Complement- and macrophage-mediated cytotoxicity [41].
Trastuzumab-emtansine	Taken up by endosomes and degraded, releasing T-DM1 Blocks the HER-2 signaling pathway and mediates ADCC	DC maturation and production of proinflammatory cytokines by ansamitocin P3. Enhanced antigen uptake and activation of tumor-specific T cells in tumor-draining lymph nodes [42,43]. Massive infiltration of T cells, polarization of Th1 cells, and increases in Tregs in combination with anti-CTLA-4/PD-1 agents [43].

Abbreviations: ICD = immunogenic cell death; IFN-γ = interferon γ; MDSC = myeloid-derived suppressor cell; NK = natural killer; Treg = regulatory T cell; AC = doxorubicin and cyclophosphamide; Tc1 = type 1 immune response; TIL = tumor-infiltrating lymphocyte; GM-CSF = granulocyte macrophage colony-stimulating factor; pNK = peripheral NK cell activity; TLR4 = toll-like receptor 4; STAT3 = signal transducer and activator of transcription 3; HER-2 = human epidermal growth factor receptor 2; ADCC = antibody-dependent cellular cytotoxicity; T-DM1 = trastuzumab-emtansine; Th1 = T helper 1; CTLA-4 = cytotoxic T lymphocyte antigen 4; PD-1 = programmed death 1.

## Data Availability

Not applicable.

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
