# Peer review of "Current and Future Therapies for Immunogenic Cell Death and Related Molecules to Potentially Cure Primary Breast Cancer"

_cancers, 2021, doi:10.3390/cancers13194756_

Round 1

Reviewer 1 Report

Authors represented the review paper entitled Current and Future Therapies for Immunogenic Cell Death and Related Molecules to Cure Primary Breast Cancer. The manuscript is well presented and structured.

Some minor spell and grammar checks are required, e.g. line 25 p.1 modulations instead of modulation.

Author Response

We are grateful to the reviewers for their careful appraisal of our manuscript. Our responses to the points raised are provided below.

Reviewer 1

In response to the reviewer’s comment, we have proofread the manuscript and resolved all minor spelling and grammatical issues (e.g., P1, L17).

Reviewer 2 Report

Still concerned about using the words “to cure” in title. These words should either be removed or changed to “potentially cure”. It is misleading when you say "to cure"

Line 27: This response eradicates – change to “can possibly eradicate”

Line 56: Should be referenced 5 AND 9

Line 678: This section still needs a title, example “Future Therapies”

Author Response

We are grateful to the reviewers for their careful appraisal of our manuscript. Our responses to the points raised are provided below.

Reviewer 2

  1. At the reviewer’s suggestion, we have changed “to cure” to “to potentially cure” in the title.
  2. We have replaced “eradicates” with “may eradicate” (P2, L19).
  3. Reference #9 (renumbered to #6) has been added along with reference #5 (P3, L15).
  4. At the reviewer’s suggestion, we have added the section heading “Future Therapies” (P17, L27).

This manuscript is a resubmission of an earlier submission. The following is a list of the peer review reports and author responses from that submission.

Round 1

Reviewer 1 Report

The review entitled "The Role of Immunogenic Cell Death and Related Molecules in Treatment to Cure Primary Breast Cancer" by Dr. Kim and Dr. Kin is a very interesting review on cancer cell death by immunogenic or nonimmunogenic stimulus framed in BC anticancer therapies.

The present review is focused on damage-associated molecular patterns (DAMPs) and activation of host immune system with many figures to improve clarity.
However, the abstract should be more focused on the aims and final goal of the review. Various cell death effects during anticancer agents-mediated BC cell death are included in the manuscript. BC is a largely heterogeneous disease with distinct molecular subtypes associated with distinct outcomes.
The narrative synthesis is fine but I would encourage to summarize anticancer drugs in a table including an overview of BC guideline recommendations. Moreover, could be useful for readers, summarize other drugs, whose anticancer mechanisms need to be furtherly elucidated. 
For instance, recently, Alisertib, a selective Aurora kinase A (AURKA) inhibitor, showed apoptotic effects on distinct BC cells, please cite (https://www.mdpi.com/1422-0067/21/18/6690/htm).

Overall, the review is well-written and well organized with minor English spell checks.

Reviewer 2 Report

Authors presented the manuscript entitled The Role of Immunogenic Cell Death and Related Molecules in
Treatment to Cure Primary Breast Cancer.

Please provide more information dealing with checkpoint inhibitors. I suggest adding several phrases on targeting different time points to increase the therapeutic potency of the chosen drug.

Reviewer 3 Report

Reference # 2 and # 3 should be reversed, information from #2 is referenced as #3 in article

Reference # 5 Does not mention Annexin A1, needs reference

Reference # 6 Lines 48 to 51 need a reference - #6 is not correct reference

Reference # 7 should be #6 – information from #7 is referenced as #6

Lines 61 to 63 should be referenced as #8

# 14 Only partial information in article (toll-like receptor mentioned) – other information not mentioned in this article (Lines 112-115)

Section 2 of article talks about select anticancer agents and their effects on immune system. It may be worthwhile exploring some other commonly used agents as well: Nab-paclitaxel in section 2.2; Methotrexate in section 2.3; Capecitabine in section 2.4; pertuzumab and trastuzumab-emtansine in section 2.5. Reference #89 has some of this information

Reference # 15 Does not mention that epirubicin is favourable in breast cancer – doesn’t even mention epirubicin by name, how did they extrapolate this information?

Reference # 16 Does not talk about effect of doxorubicin on myeloid-derived suppressor cells. This info is in #17 – not referenced correctly

Lines 136 – 142 are all from #17, referenced incorrectly to # 16

Line 161 – should say “based” not “biased”

Line 181-182 – statement is INCORRECT – Article talks about doxorubicin/cyclophosphamide combination therapy – cannot say that cyclophosphamide alone increases number of MDSCs

Reference # 34 & # 35 are the EXACT same article with different authors? Published in the same journal?

Lines 240 – 248 do not have a proper reference, some info from #41 but not all of it

Line 249 is from # 40

Line 370 should either say “normal allele counterparts” or “as compared to patients with normal alleles”

Line 428 should be referenced to both #82 and #83

Line 430 should be referenced to #83

Line 469 – should be IL-1 instead of IL-10??

Reference #94 Has all the info required for lines 530-532. #95 is unnecessary

Line 559 – no section heading and starts to talk about a treatment that was not mentioned anywhere else in the article prior. Can be titled – current and future treatment options

Article mentions vaccine but does not talk about the vaccine developed already for breast cancer (https://www.hopkinsmedicine.org/kimmel_cancer_center/cancers_we_treat/breast_cancer_program/_archive/breast_cancer_vaccine.html)